# THIS LOOKS LIKE IT RATHER THAN THAT: PROTOKNN FOR SIMILARITY-BASED CLASSIFIERS

**Yuki Ukai**[*1,2], **Tsubasa Hirakawa**[2], **Takayoshi Yamashita**[2], **and Hironobu Fujiyoshi**[2]
[1]GLORY Ltd., [2]Chubu University

## ABSTRACT

Among research on the interpretability of deep learning models, the 'this looks like that' framework with ProtoPNet has attracted significant attention. By combining the strong power of deep learning models with the interpretability of case-based inference, ProtoPNet can achieve high accuracy while keeping its reasoning process interpretable. Many methods based on ProtoPNet have emerged to take advantage of this benefit, but despite their practical usefulness, they run into difficulty when utilizing similarity-based classifiers. This is because ProtoPNet and its variants adopt the training process specific to linear classifiers, which allows the prototypes to represent useful image features for class recognition. Due to this difficulty, the effectiveness of similarity-based classifiers (e.g., k-nearest neighbor (KNN)) on the 'this looks like that' framework have not been sufficiently examined. To alleviate this problem, we propose ProtoKNN, an extension of ProtoPNet that adopts KNN classifiers. Extensive experiments on multiple open datasets demonstrate that the proposed method can achieve competitive results with a state-of-the-art method.

## 1 INTRODUCTION

Deep learning has achieved very high accuracy in a variety of computer vision tasks. However, since the reasoning process of deep learning models is black-boxed and cannot be interpreted by human operators, it is very difficult to validate their inference, and this impedes their utilization in high-risk domains. To alleviate this problem, several methods for constructing inherently interpretable models have been proposed. However, inherently interpretable models generally suffer from degraded accuracy compared to black-box models. 'Gray-box' models have thus been proposed (Alvarez-Melis, 2018; Chen, 2019; Koh, 2020) to take advantage of the power of deep learning models while keeping the reasoning process interpretable. Among the gray-box model approaches, the 'this looks like that' framework with ProtoPNet (Chen, 2019) has attracted significant attention because it can guarantee a transparent reasoning process without any additional supervision. ProtoPNet first calculates the similarity of the input samples to the prototypes corresponding to an image patch in the training set and then classifies samples with inherently interpretable models on the basis of this similarity. This process enables ProtoPNet to explain its reasoning process by providing patches in the training set that the model considers similar to the input sample. Thus, interpretability with case-based reasoning is achieved. Thanks to this advantage in transparency, many methods based on ProtoPNet have been proposed (Wang, 2021; Nauta, 2021; Rymarczyk, 2021; Donnelly, 2022; Keswani, 2022; Rymarczyk, 2022).

When training ProtoPNet, the weights of the linear classifier connecting each of the prototypes and class logits are fixed, and the feature vectors corresponding to an image patch are linked to the prototypes if the prototypes make a positive contribution to the class logits to which the image belongs. This enables the prototypes to represent the image patches most useful for class recognition. However, due to this special training process, it is difficult for ProtoPNet to utilize any classifiers other than the linear classifier. As an alternative, Nauta (2021) proposed ProtoTree, which use a decision tree for the last classifier. However, this method is limited to the decision tree, which makes it difficult to utilize in similarity-based classifiers with the 'this looks like that' framework. Similarity-based classifiers perform inference on the basis of similarities (or distances) between samples. As we will demonstrate in the experimental section and in the Appendix (Sec. D.2), interpreting the distance enables us to obtain more fine-grained explanation in a counterfactual manner, which is

important for understanding and interpreting the reasoning process of the model. Therefore, in this work, we extend ProtoPNet so that we can utilize similarity-based classifiers (specifically, k-nearest neighbor (KNN) classifiers) in the 'this looks like that' framework.

When extending ProtoPNet to the similarity-based classifier, it is no longer possible to pre-define the relationship between the prototypes and the class labels, which is necessary for calculating the cluster loss. As we will discuss in Sec. 2.2.2, it is also difficult to estimate this relationship from only one sample. Therefore, the main difficulty is how to estimate the relationship between the prototypes and the class labels. Our concept for estimating this relationship is to compare each sample in a mini-batch and sum up the most distinctive prototypes. This enables us to estimate which prototypes are relatively more related to which samples and thus which classes. Then, our novel cluster loss can be defined based on this estimation. In summary, our contributions are three-fold:

- We propose ProtoKNN, an extension of ProtoPNet that can utilize KNN classifiers. This is the first work to examine the effectiveness of similarity-based classifiers in the 'this looks like that' framework.
- We developed a novel loss function for ProtoKNN that replaces the cluster loss in ProtoPNet. This enables us to train our model without predefining the relationship between classes and prototypes.
- The proposed method achieved competitive results with a state-of-the-art ProtoPNet variant on multiple open datasets.

## 2 METHOD

In the following, we first describe the notation [1] used in this paper. Then, we present the training strategy of the proposed method and explain how to classify the samples. Finally, we demonstrate how to interpret the reasoning processes of our method. For context, we also briefly revisit the origins of ProtoPNet and elaborate on why it cannot directly utilize similarity-based classifiers in the Appendix (Sec. A).

### 2.1 PRELIMINARY

The input image and its class labels are denoted as $x$ and $y$, respectively. Unless otherwise specified, we use subscripts $a, b, ...$ to denote the data indices. Thus, an input image and its class label are denoted as $x_a$ and $y_a$, respectively. The index sets of the images in a minibatch and their cardinality are denoted as $\mathbb{B}$ and $|\mathbb{B}|$, respectively. We use $F$ to denote the feature extractor and $\mathbf{Z}$ to denote the feature map output by $F$, i.e., $\mathbf{Z}_a = F(x_a)$. $z$ is used to denote the feature vectors contained in a pixel of the feature map $\mathbf{Z}$. We also call these feature vectors 'image patch features' in this paper. After the transformation, the similarity between the prototypes $\{\boldsymbol{p}_i\}_{i=0,1,...}$ and the image patch features contained in the feature map $\mathbf{Z}_a$ are calculated. The maximum similarity value is defined as the similarity of the input image $x_a$ to the prototype $\boldsymbol{p}_i$, as $s_{a,i} = \max_{\boldsymbol{z} \in \mathbf{Z}_a} Sim(\boldsymbol{z}, \boldsymbol{p}_i)$. Here, we denote the similarity of the input image $x_a$ as $\boldsymbol{s}_a$ and its component corresponding to $\boldsymbol{p}_i$ as $s_{a,i}$. $Sim$ is the function that calculates the similarity (cosine similarity in this paper) between the feature vectors and the prototypes. In the following, we refer to the similarity $\boldsymbol{s}_a$ as the 'prototype profile' of the input image $x_a$, and the index sets of the prototypes are denoted as $\mathbb{P}$. The indicator function is denoted as $\mathbf{1}(\text{condition})$, which returns 1 if the condition is true and 0 otherwise.

### 2.2 TRAINING PROCESS

Originally, ProtoPNet utilized three loss functions: classification loss, cluster loss, and separation loss. In the proposed method, we do not use the separation loss because we expect the prototypes to be common among the samples with different class labels. Instead, we use the auxiliary loss function proposed in the context of deep metric learning to help the backbone model acquire better feature extraction ability. In summary, we train our models with three loss functions: classification loss $L_{task}$, the novel cluster loss $L_{clst}$, and auxiliary loss $L_{aux}$. Figure 1 shows the loss scheme of the proposed method. The details of each loss are described in the following.

---

[1]We basically follow `https://github.com/goodfeli/dlbook_notation/`.

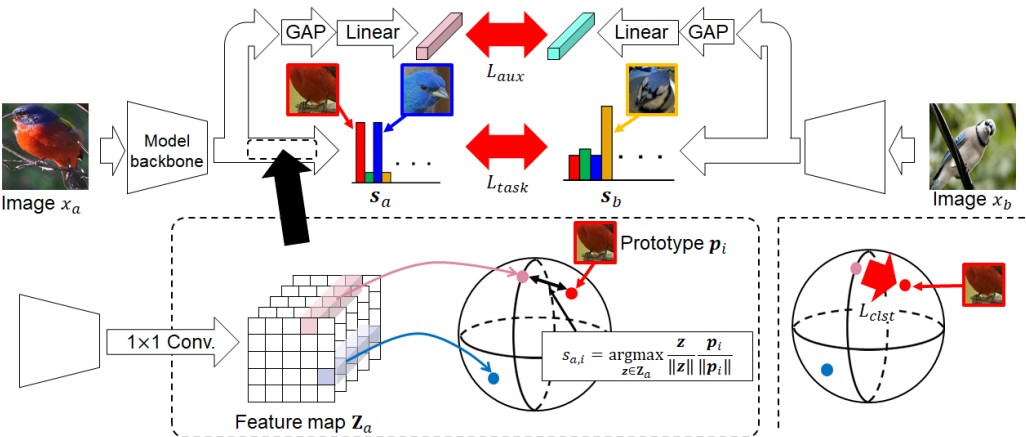

Figure 1: The loss scheme of the proposed method. The model transforms the input sample $x_a$ into the feature map $\mathbf{Z}_a$ and then encode it into the prototype profile $s_a$. We measure the distance between images on the space spanned by $s_a$. To optimize the distance, we impose $L_{task}$ on $s_a$. We also impose $L_{clst}$ to link the prototypes and the feature vectors contained in $\mathbf{Z}_a$. In addition, to help the model acquire the capability of feature extraction, we directly impose $L_{aux}$ on the output of the model backbone.

### 2.2.1 CLASSIFICATION LOSS $L_{task}$

In contrast to ProtoPNet and its variants, the proposed method performs inference on the basis of similarity between samples. Therefore, in the space spanned by the similarity between the prototypes and the input images, it needs to be 'discriminable' rather than 'separable'. We therefore use the loss function (Wu, 2017) typically utilized in the context of deep metric learning instead of the ordinarily used cross-entropy loss for $L_{task}$. Thus, we calculate $L_{task}$ as

$$L_{task} = \frac{1}{N} \sum_{a \in B} ([d_{ap} + m - \beta]_+ + [m - d_{an} + \beta]_+), \qquad (1)$$

where $m$, $\beta$ are margin and learnable parameters, and $a$, $p$, and $n$ are the data indices of the anchor, the hard-positive, and the hard-negative samples, respectively. The ReLU function is denoted as $[\cdot]_+$, and $N$ is the number of terms that take a non-zero value in the summation. $d_{ap}$ and $d_{an}$ are the Euclidean distance defined as $\|s_a - s_p\|_2$ and $\|s_a - s_n\|_2$, respectively.

### 2.2.2 ESTIMATION OF PROTOTYPE ATTRIBUTION

As we will discuss in the Appendix (Sec. A), the predefined relationship between the prototypes and the class labels is no longer available. Thus, to calculate the cluster loss, we need to calculate which image patch features should be attached to each of the prototypes and to what degree. For this purpose, we first estimate the affiliation of the prototypes to each class.

Specifically, we first estimate which prototypes are contained in which samples. Utilizing the prototype profiles would be a natural choice here, but, thresholding methods are not appropriate because the prototype profiles change their statistical properties during the training. Therefore, we estimate the affiliation by comparing the prototype profiles, i.e., the similarity of the samples to the prototypes, in the minibatch. If the image feature represented by a prototype $p_i$ is contained in sample $x_a$ and not in sample $x_b$, we can expect that the similarity ($s_a$) of sample $x_a$ to prototype $p_i$ is greater than the similarity ($s_b$) of sample $x_b$ to prototype $p_i$. Therefore, by extracting the prototypes with a large difference between $s_a$ and $s_b$ for all of the indices in the minibatch ($b \in \mathbb{B}$), we can construct a set of the prototypes contained in $x_a$. To achieve this, stochastic sampling with the Gumbel-Max trick is conducted by

$$E(x_a, \boldsymbol{p}_i) = \sum_{b \in \mathbb{B}/\{a\}} \Gamma\left(\frac{s_{a,i} - s_{b,i}}{\tau}\right), \qquad (2)$$

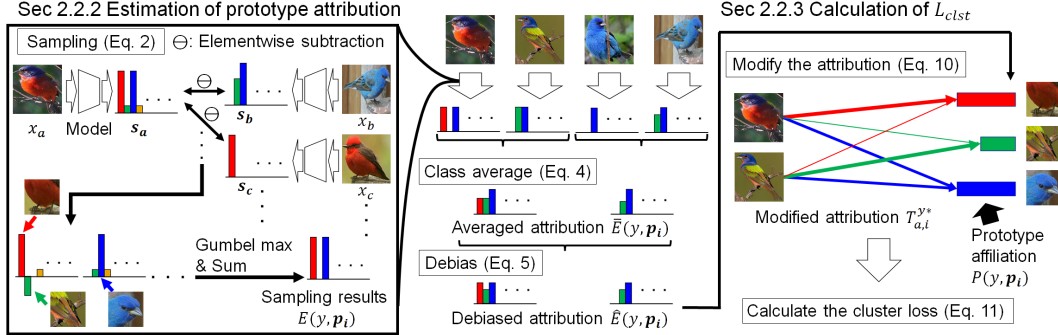

Figure 2: Overall process of calculating the cluster loss. First, the prototype profiles are compared to estimate which prototypes are likely to be included in each sample (this process is denoted as 'Sampling'). Second, we average the 'sampling results' within a class and then debias the results so that the class average of all results is the same. Third, the debiased results are normalized so that the total amount of the prototypes contained in a class is 1. We refer to the normalized results as 'prototype affiliation'. Finally, we modify the prototype affiliation and link the prototypes to the image patch features on the basis of the modified attribution. Please refer to the main text for the details of each process.

where $\tau$ is the temperature parameter (fixed to $0.05$ in this paper). $\Gamma$ is the Gumbel-Max operation (Jang, 2017) and returns 1 for the index of the sampled prototype and 0 otherwise, i.e.,

$$\Gamma(s_i) = \mathbf{1}\left(i = \arg\max_j \frac{\exp(s_j + \gamma_j)}{\sum_k \exp(s_k + \gamma_k)}\right),\tag{3}$$

where $\gamma_j$ and $\gamma_k$ are the random variables that follow the standard Gumbel distribution. We obtain the sampling results $\bar{E}(y, p_i)$ with respect to each class label $y$ by averaging the sampling results $E(x_a, p_i)$ among the samples that have the same class labels, as

$$\bar{E}(y, \boldsymbol{p}_i) = \frac{\sum_{a \in \mathbb{B}} E(x_a, \boldsymbol{p}_i)\mathbf{1}(y_a = y)}{\sum_{b \in \mathbb{B}} \mathbf{1}(y_b = y)}.\tag{4}$$

The sampling results $\bar{E}(y, p_i)$ can still be biased toward some prototypes independent of the class labels, which is undesirable. Thus, we de-bias the sampling results following the equation below so that they are not biased toward any specific prototype.

$$\hat{E}(y, \boldsymbol{p}_i) = \frac{\bar{E}(y, \boldsymbol{p}_i)}{\sum_{y_a \in \mathbb{Y}} \bar{E}(y_a, \boldsymbol{p}_i)}\tag{5}$$

where $\mathbb{Y}$ is the set of class labels contained in the mini-batch. This 'debiasing' enforces the sum of the sampling results with respect to the class labels are the same. In other words, the debiasing process reflects the assumption that in the absence of information about the class, the same amount of prototypes are observed with probability. Finally, we calculate the affiliation of the prototypes to the class label $y$ by normalizing the sampling results $\hat{E}(y, p_i)$ so that the total amount of the prototypes contained in a class would be the same (i.e., 1), as

$$P(y, \boldsymbol{p}_i) = \frac{\hat{E}(y, \boldsymbol{p}_i)}{\sum_{j \in \mathbb{P}} \hat{E}(y, \boldsymbol{p}_j)}\tag{6}$$

### 2.2.3 CLUSTER LOSS $L_{clst}$

Once we estimate the prototype affiliation (Sec. 2.2.2), we can calculate and minimize the cluster loss to link the image patch features to the prototypes on the basis of that estimation. Here, due to the various views of objects in the images, the samples do not always contain the prototypes whose affiliation to the class is high. Therefore, we need to modify the affiliation for each sample. Note that we can relatively estimate how much the sample $x_a$ contains the image feature represented by

the prototype $\boldsymbol{p}_i$ on the basis of the distance between them $(\min_{\boldsymbol{z} \in \mathbf{Z}_a} \|\boldsymbol{z} - \boldsymbol{p}_i\|_2^2)$. Therefore, we propose to modify the attribution considering the distance. Specifically, we formulate this problem as a linear assignment problem:

$$T_{a,i}^{y*} = \arg\min_{T_{a,i}^y} \sum_{a \in \mathbb{B}, i \in \mathbb{P}} T_{a,i}^y C_{a,i} \ s.t. \sum_{i \in \mathbb{P}} T_{a,i}^y = \mathbf{1}(y_a = y), \sum_{a \in \mathbb{B}} T_{a,i}^y = N_y P(y, \boldsymbol{p}_i), \quad (7)$$

where $N_y = \sum_b \mathbf{1}(y_b = y)$ is the number of samples contained in a mini-batch whose class label is $y$ and $C_{a,i}$ is the cost function defined as

$$C_{a,i} = \min_{\boldsymbol{z} \in \mathbf{Z}_a} \|\boldsymbol{z} - \boldsymbol{p}_i\|_2^2. \quad (8)$$

The solution $T_{a,i}^{y*}$ can be considered as the modified attribution of the prototypes to each sample, whose average among the samples match the attribution to the class. The components of $T_{a,i}^{y*}$ would be small when the sample $x_a$ does not contain the prototype $\boldsymbol{p}_i$, i.e., when the distance between them is large, and high the other way around. Thus, we use the solution $T_{a,i}^{y*}$ to calculate the cluster loss, as

$$L_{clst} = \sum_{y \in \mathbb{Y}} \sum_{a \in \mathbb{B}, i \in \mathbb{P}} T_{a,i}^{y*} C_{a,i}. \quad (9)$$

Here, $\mathbb{Y}$ is the set of all class labels contained in the minibatch.

To minimize Eq. 9, we need to solve Eq. 7. However, quickly obtaining an exact solution to Eq. 7 is difficult, and it is not necessary for it to be unique. Therefore, we approximately solve Eq. 7 with the Sinkhorn-Knopp algorithm (Cuturi, 2013) and re-define $T_{a,i}^{y*}$ as

$$\begin{aligned} T_{a,i}^{y*} &= \arg\min_{T_{a,i}^y} \sum_{a \in \mathbb{B}, i \in \mathbb{P}} T_{a,i}^y C_{a,i} + \frac{1}{\lambda} T_{a,i}^y \log T_{a,i}^y \\ s.t. &\sum_{i \in \mathbb{P}} T_{a,i}^y = \mathbf{1}(y_a = y), \sum_{a \in \mathbb{B}} T_{a,i}^y = N_y P(y, \boldsymbol{p}_i) \end{aligned} \quad (10)$$

where $\lambda$ is the hyper-parameter determining the weight of the entropy regularization term (fixed to 0.05 in this paper).

We also modified the proposed cluster loss to maximize the gap between the minimum and the average distance between the prototypes and the image patch features following ProtoPool(Rymarczyk, 2022). The aim of this modification is to address the problem that the learned prototypes tend to focus on the background (Rymarczyk, 2022). We re-define our cluster loss (Eq. 9) as

$$L_{clst} = \sum_{y \in \mathbb{Y}} \sum_{a \in \mathbb{B}, i \in \mathbb{P}} T_{a,i}^{y*} \hat{C}_{a,i}, \ where \ \hat{C}_{a,i} = \min_{\boldsymbol{z} \in \mathbf{Z}_a} \|\boldsymbol{z} - \boldsymbol{p}_i\|_2^2 - \frac{1}{|\mathbf{Z}_a|} \sum_{\boldsymbol{z} \in \mathbf{Z}_a} \|\boldsymbol{z} - \boldsymbol{p}_i\|_2^2 \quad (11)$$

where $|\mathbf{Z}_a|$ is the number of pixels contained in the feature map $\mathbf{Z}_a$.

### 2.2.4 AUXILIARY LOSS $L_{aux}$

As stated earlier, we utilize the loss function proposed in the context of deep metric learning as auxiliary loss $L_{aux}$ to help the model backbone acquire better feature extraction ability. Here, we also specify that we do not impose $L_{aux}$ on the prototype profile $\boldsymbol{s}$ but on the feature vectors obtained by applying Global Average Pooling (GAP) and a linear layer to the feature map output by the model backbone. We refer to this feature vector and its L2-normalized one as $\boldsymbol{g}$ and $\hat{\boldsymbol{g}}$ in this section. As a default, we used proxy anchor loss (Kim, 2020) for $L_{aux}$, which is formulated as:

$$L_{aux} = \frac{1}{|Q^+|} \sum_{\boldsymbol{q} \in Q^+} \log\left(1 + \sum_{\hat{\boldsymbol{g}} \in X_p^+} e^{-\alpha(s(\hat{\boldsymbol{g}}, \boldsymbol{q}) - \delta)}\right) + \frac{1}{|Q|} \sum_{\boldsymbol{q} \in Q} \log\left(1 + \sum_{\hat{\boldsymbol{g}} \in X_p^-} e^{\alpha(s(\hat{\boldsymbol{g}}, \boldsymbol{q}) - \delta)}\right) \quad (12)$$

where $\delta$, $\alpha$, $Q$, and $Q^+$ are margin, scaling factor, the set of all proxies, and the set of positive proxies of data in the mini-batch, respectively. Also, for each proxy $\boldsymbol{q}$, the batch of feature vectors is divided into two sets: $X_p^+$, a set of positive feature vectors of $\boldsymbol{q}$ and $X_p^-$, the others. We also defined $s(\hat{\boldsymbol{g}}, \boldsymbol{q}) = \hat{\boldsymbol{g}} \frac{\boldsymbol{q}}{\|\boldsymbol{q}\|_2}$ above.

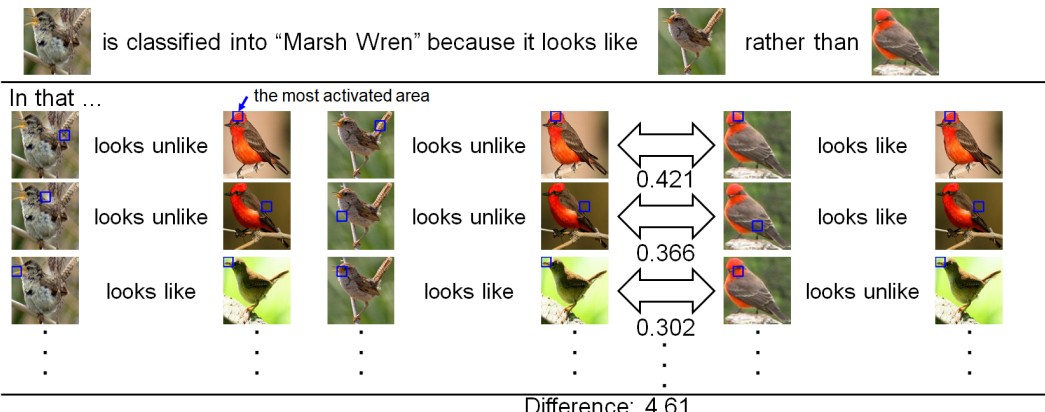

Figure 3: An example of the interpretation of the proposed method. Blue bounding boxes surround the areas where the prototypes are the most activated. The numerical values in the center of the figure indicate how much the distance between the anchor and the negative sample was increased by the corresponding prototype relative to the distance between the anchor and the positive sample. Please refer to the final paragraph of Sec. 2.4 for more details.

### 2.2.5 OVERALL LOSS FUNCTION AND PROTOTYPE PROJECTION

We train our models with the combination of the loss functions described above. Specifically, the overall loss function of the proposed method can be written as

$$L_{total} = L_{task} + \lambda_{clst} \cdot L_{clst} + \lambda_{aux} \cdot L_{aux}, \tag{13}$$

where $\lambda_{clst}$ and $\lambda_{aux}$ are the hyper-parameters that respectively determine the weight of each loss. We empirically set $\lambda_{clst} = 0.1$ and $\lambda_{aux} = 1.0$ in this paper. After the training, each prototype is projected onto the nearest image patch features in the training set following

$$\boldsymbol{p}_i \leftarrow \arg\max_{\boldsymbol{z}} \frac{\boldsymbol{z}}{\|\boldsymbol{z}\|} \frac{\boldsymbol{p}_i}{\|\boldsymbol{p}_i\|} \tag{14}$$

This guarantees that each prototype represents the corresponding image patches, and thus the transparency of the prototypes is achieved. We can then achieve case-based interpretability by classifying samples on the basis of their similarity to the prototypes.

### 2.3 LABEL PREDICTION

To predict the class label with our method, given the test sample $x_{test}$, we first retrieve the nearest top-k samples from the training set. Here, k is the hyper-parameter of the KNN classifier and we use the Euclidean distance for the distance metric, i.e., the difference between samples $x_a$ and $x_b$ is calculated by $\|\boldsymbol{s}_a - \boldsymbol{s}_b\|_2^2$. Then, the prediction $y_{pred}$ is given by the majority decision of the retrieved top-k samples $R(x_{test}, k)$, i.e., $y_{pred} = \arg\max_y \sum_{x_a \in R(x_{test}, k)} \mathbf{1}(y_a = y)$. In the experimental section, we mainly report the results with $k = 1, 3$, and $5$.

### 2.4 INTERPRETATION OF THE REASONING PROCESS

We limit the following explanation of the proposed method's inference interpretation to the case of $k = 1$ for simplicity. The proposed method can achieve a counterfactual explanation for why the input sample is classified into the predicted class rather than another class. In the following, we refer to the input image as the anchor sample and to the sample nearest to the anchor sample in the training set as a positive sample. We also refer to the class of interest other than the predicted class as a negative class and the nearest sample with the negative class label to the anchor sample in the training set as a negative sample.

As shown in Fig. 3, the proposed method explains its inference by using the positive and negative samples. In contrast to the naive KNN classifier, which only provides the positive sample for the

Table 1: Accuracy comparison on full images of CUB200-2011 with various model backbones. The results of ProtoPNet are directly borrowed from Donnelly (2022). 'V', 'R', and 'D' denote VGG, ResNet, and DenseNet, respectively. Here, the top-1 accuracy [%] is reported.

| Model | V16 | V19 | R34 | R50 | R152 | D121 | D161 |
|---|---|---|---|---|---|---|---|
| ProtoPNet(Chen, 2019) | 70.3 | 72.6 | 72.4 | 81.1 | 74.3 | 74.0 | 75.4 |
| Def. ProtoPNet(Donnelly, 2022) | 75.7 | 76.0 | 76.8 | 86.4 | 79.6 | 79.0 | 81.2 |
| ProtoKNN (k=1) + Margin | 75.5 | 76.7 | 76.9 | 87.0 | 80.3 | 78.5 | 80.8 |
| ProtoKNN (k=3) + Margin | 76.6 | 77.2 | 77.3 | 87.1 | **80.6** | 79.3 | 81.2 |
| ProtoKNN (k=5) + Margin | 77.2 | **77.6** | **77.6** | **87.2** | **80.6** | 79.8 | **81.4** |
| ProtoKNN (k=10) + Margin | **77.5** | **77.6** | 77.4 | 87.0 | 80.5 | **79.9** | 81.3 |

reason of its inference, the proposed method can further explain why the distance between the anchor and the negative samples is larger than the distance between the anchor and the positive samples. Later, we will refer to the distance between the anchor and the positive (negative) samples as $d_{ap}$ ($d_{an}$). Here, we used the subscripts $a$, $p$, and $n$ to denote the data indices of the anchor, the positive, and the negative samples, respectively. In the proposed method, the distance between two samples is determined by the difference between the similarity of each sample to the prototypes. Therefore, by extracting the prototypes that have a large contribution to $d_{an}$ and a small contribution to $d_{ap}$, we can explain which image feature is responsible for the difference between $d_{ap}$ and $d_{an}$. In other words, the prototype with the largest contribution to $d_{an}^2 - d_{ap}^2 = \|s_a - s_n\|_2^2 - \|s_a - s_p\|_2^2$ can be considered as the reason for why the anchor sample is classified into the predicted class.

Consider the interpretation example shown in Fig. 3. First, the anchor sample (top left) and the negative class (Vermilion Flycatcher) are given. Then, the positive (top center) and the negative (top right) samples are retrieved from the training dataset. As a result, we find that $d_{an}^2$ is 4.61 greater than $d_{ap}^2$ (denoted as 'Difference') in this case. Next, as described above, we extract the prototypes that make the largest contribution to $d_{an}^2 - d_{ap}^2$, as shown in the middle table in Fig. 3. The extracted results are the reason for why the anchor sample is classified into 'Marsh Wren' rather than 'Vermilion Flycatcher'. Specifically, these results show that the main reason is that the negative sample has a red head, while the anchor and the positive samples do not. We can also see that this difference makes $d_{an}^2$ 0.421 greater than $d_{ap}^2$. Similarly, by repeating the interpretation described above for the second and third results, we can quantitatively interpret why $d_{an}^2$ is greater than $d_{ap}^2$. Further examples for the interpretation of the proposed method can be found in the Appendix.

## 3 EXPERIMENTAL RESULTS

We conducted experiments on three public datasets: CUB200-2011 (Wah, 2011), Stanford Dogs (Khosla, 2011), and Stanford Cars (Krause, 2013). Due to space limitations, we only describe the experimental results here. Other details (e.g., implementation details) are provided in the Appendix.

### 3.1 COMPARISON WITH THE OTHER METHOD

Tables 1 and 2 show the top-1 accuracy of the proposed method with various model backbones on CUB200-2011 and Stanford Dogs, respectively. Here, we directly borrowed the results of ProtoPNet from Donnelly (2022). In these experiments, we used the full images following Donnelly (2022). As shown, our method achieved competitive results with Deformable ProtoPNet (Donnelly, 2022).

Tables 3 and 4 show the comparison results with the state-of-the-art method among variants of ProtoPNet. Here, we cropped the images using bounding boxes following the previous studies (Chen, 2019; Wang, 2021; Rymarczyk, 2022). We also set the number of prototypes to that of ProtoPool (Rymarczyk, 2022). As we can see, our method achieved a higher accuracy than the other method when using a single model, and it even achieved competitive results with the ensemble results of the other method in Table 4. Specifically, our method achieved a higher accuracy than ProtoPool when we used the same model backbone and the same number of prototypes. These findings demonstrate the effectiveness of our method, which eliminates the need for the sub-optimal optimization process with a fixed-weight classification layer required by the conventional methods.

Table 2: Accuracy comparison on full images of Stanford Dogs with various model backbones. 'V', 'R', and 'D' denote VGG, ResNet, and DenseNet, respectively. Here, the top-1 accuracy [%] is reported.

| Method | V19 | R152 | D161 |
|---|---|---|---|
| ProtoPNet(Chen, 2019) | 73.6 | 76.2 | 77.3 |
| Def. ProtoPNet(Donnelly, 2022) | 77.9 | 86.5 | 83.7 |
| ProtoKNN (k=1) + Margin | 76.9 | 85.9 | 84.7 |
| ProtoKNN (k=3) + Margin | 78.6 | 86.9 | **85.7** |
| ProtoKNN (k=5) + Margin | 79.3 | 87.3 | **85.7** |
| ProtoKNN (k=10) + Margin | **79.5** | **87.5** | 85.7 |

Table 3: Accuracy comparison on Stanford Cars with state-of-the-art method. Here, 'R' denotes ResNet and top-1 accuracy [%] is reported.

| Method | R50 |
|---|---|
| ProtoTree(Nauta, 2021) | 86.6 |
| ProtoPool(Rymarczyk, 2022) | 88.9 |
| ProtoKNN (k=1) + Margin | 90.2 |
| ProtoKNN (k=3) + Margin | 90.8 |
| ProtoKNN (k=5) + Margin | **90.9** |
| ProtoKNN (k=10) + Margin | 90.7 |

Table 4: Accuracy comparison on CUB200-2011 with state-of-the-art method. '*' and '**' denote the ensemble results with 3 and 5 models, respectively.

| Method | Top-1 Acc. |
|---|---|
| ProtoPNet(Chen, 2019) | 79.2% (*84.8%) |
| ProtoTree(Nauta, 2021) | 82.2% (**87.2%) |
| TesNet(Wang, 2021) | 82.8% (**86.2%) |
| ProtoPool(Rymarczyk, 2022) | 85.5% (**87.6%) |
| Def. ProtoPNet(Donnelly, 2022) | 86.4% (**87.8%) |
| ProtoKNN (k=1) + Margin | 87.1% |
| ProtoKNN (k=3) + Margin | 87.4% |
| ProtoKNN (k=5) + Margin | **87.5%** |
| ProtoKNN (k=10) + Margin | 87.2% |

Table 5: Ablation study on $L_{aux}$ using full images of CUB200-2011. Top-1 accuracy [%] is reported here.

| k | $L_{aux}$ | Res34 | Res50 |
|---|---|---|---|
| k=1 | None | 74.7% | 86.0% |
| k=3 | None | 75.3% | 86.5% |
| k=5 | None | 75.9% | 86.6% |
| k=1 | Margin | 76.9% | 87.0% |
| k=3 | Margin | 77.3% | 87.1% |
| k=5 | Margin | **77.6%** | 87.2% |
| k=1 | Proxy-Anchor | 77.0% | 87.0% |
| k=3 | Proxy-Anchor | 77.5% | **87.3%** |
| k=5 | Proxy-Anchor | 77.3% | 87.2% |

## 3.2 ABLATION STUDY

To examine the effectiveness of each component of the proposed method, we conducted an ablation study on CUB200-2011 with the full images. In the following, we present the ablation study for $L_{aux}$. We include the ablation study for our cluster loss $L_{clst}$ in the Appendix.

The results of the ablation study are shown in Table 5. As we can see, the accuracy improved regardless of whether margin loss or proxy anchor loss was adopted for $L_{aux}$. Thus, we can confirm that $L_{aux}$ helps the backbone model to acquire better feature extraction ability. We can also see that the proposed method utilized the learned good feature representation whether the ranking-based or classification-based loss was adopted for $L_{aux}$. Moreover, the improvement was marginal when ResNet50 was adopted for the model backbone compared to when ResNet34 was adopted. This is because we used ResNet50 pretrained on i-Naturalist 2017, so it has already acquired good feature extraction ability. Therefore, we can conclude that $L_{aux}$ is more effective when the pretrained model has not already acquired good feature extraction ability.

## 4 RELATED WORKS

Research on the interpretability of deep learning models can be divided into two approaches: (1) the post-hoc approach, which analyzes trained black-box models, and (2) the ante-hoc approach, which constructs inherently interpretable models. The post-hoc approach (Zhou, 2016; Selvaraju, 2017; Lundberg, 2017; Bau, 2017; Jiang, 2021; Hernandez, 2022) is beneficial when analyzing already deployed models because no re-training is required. However, in some cases, the explanations provided by a post-hoc method will have nothing to do with the model inference (Rudin, 2019). To address this problem in fidelity, some works have utilized the Shapley value (Shapley, 1951) and proposed methods with a theoretical guarantee (Lundberg, 2017; Hamilton, 2022). However, calculating the Shapley value is NP-hard and it is difficult to apply these methods to high-dimensional

data such as raw images. Moreover, most methods explain the model inference by providing the image region that contributes the most to the model inference, and the properties of the explanation are thus very different from those provided by ProtoPNet and its variants. This makes it extremely difficult to evaluate the interpretability of these approaches in a unified manner, which is why we have omitted any comparison with the post-hoc approaches from the paper. In the following, we describe the variants of ProtoPNet (Chen, 2019), which is the basis of our method.

After Chen (2019) proposed the 'this looks like that' framework with ProtoPNet, many methods based on ProtoPNet were proposed, and they have improved in both accuracy and efficiency by reducing the number of the prototypes (Wang, 2021; Rymarczyk, 2021; Donnelly, 2022; Keswani, 2022; Rymarczyk, 2022). In particular, ProtoPool (Rymarczyk, 2022) has succeeded in directly learning the prototypes that are common among inter-class samples and achieved high accuracy. However, as we will explain in the Appendix (Sec. A), these methods all have difficulty in utilizing similarity-based classifiers. ProtoPool, even though it is quite successful, requires 'slots' specific to certain classes and still has difficulty utilizing classifiers other than linear ones. Alternatively, ProtoTree (Nauta, 2021) utilizes a decision tree for its classifier and has achieved high accuracy while reducing the number of prototypes, However, its training process is specific to the decision tree and it thus also has difficulty utilizing the similarity-based classifier. To address this problem, our work extends ProtoPNet so that we can utilize the similarity-based classifier, and to the best of our knowledge, it is the first work that combines the similarity-based classifier with the 'this looks like that' framework.

Our method is also related to research in the field of deep metric learning. Many methods have been proposed to achieve a good feature representation space in this context (Wu, 2017; Wang, 2019; Kim, 2020) , most of which utilize the cosine similarity to calculate the similarity between samples (Kim, 2020). However, it is unreasonable to L2-normalize the similarity of each sample to the prototypes, which makes it difficult to directly adopt these loss function as $L_{task}$ in our method. Instead, we adopt these losses as $L_{aux}$ to take advantage of the deep metric learning and help the model backbone acquire better feature representation.

## 5 DISCUSSION

**Limitation.** Our method utilizes the KNN classifier for classification, so we need to maintain the similarity of each sample in the training set to the prototypes during inference. We also need to calculate the distance between the input sample and each of the samples in the training dataset. This is not desirable from the viewpoint of either memory efficiency or computational efficiency. The computational cost of our method is O(ND), where N refers to the number of samples and D to the number of channel dimensions. Although this cost is negligible when adopting the datasets used in this paper, it might be problematic when scaling up to larger datasets. However, we believe these problem can be solved by properly selecting the samples from the training set to be used during the inference, which we leave to future work.

Our method also requires a sufficient number of training iterations to ensure the prototypes are close enough to certain image patch features. Since most of the deep metric learning methods tend to overfit the training dataset and degrade its accuracy the longer it is trained(Kim, 2021), we need some trick such as regularization(Kim, 2021) or distillation(Park, 2019) from a well-generalizable teacher in order to achieve high accuracy in the image retrieval settings with our method. Verifying the effectiveness of such tricks is outside the scope of this paper and we leave it to future work. Please note that the number of training iterations of our method is not higher than that of the other variants of ProtoPNet. Moreover, the training time of our method (Resnet 50 on the CUB200-2011 dataset) is nearly four hours, which is almost the same as ProtoPool in our experimental settings (one RTX3090 GPU).

**Conclusion.** In this paper, we proposed a method in which a KNN classifier is combined with the 'this looks like that' framework. Extensive experiments on multiple open datasets showed that the proposed method achieved competitive results with a state-of-the-art ProtoPNet variant, thus demonstrating the effectiveness of utilizing similarity-based classifiers in the 'this looks like that' framework.

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

## A  REVISITING PROTOPNET

In this section, we briefly revisit the origins of ProtoPNet and elaborate on why it cannot directly utilize similarity-based classifiers.

ProtoPNet first transforms an input image $x_a$ into a feature map $\mathbf{Z}_a$ with its feature extractor $F$. After the transformation, the similarity between the prototypes $\{\boldsymbol{p}_i\}_{i=0,1,\dots}$ and the image patch features contained in the feature map $\mathbf{Z}_a$ are calculated. The maximum similarity value is defined as the similarity of the input image $x_a$ to the prototypes, as

$$s_{a,i} = \max_{\boldsymbol{z}\in\mathbf{Z}_a} Sim(\boldsymbol{z}, \boldsymbol{p}_i). \tag{15}$$

Here, we denote the similarity of the input image $x_a$ as $\boldsymbol{s}_a$ and its component corresponding to $\boldsymbol{p}_i$ as $s_{a,i}$. $Sim$ is the function that calculates the similarity between the feature vectors and the prototypes. Many functional forms have been proposed for $Sim$ (Chen, 2019; Nauta, 2021; Wang, 2021; Donnelly, 2022), and in this study, we use cosine similarity. As stated in the main paper, the inference is conducted on the basis of this similarity.

The training process of ProtoPNet is divided into two steps: (1) training the feature extractor and the prototypes and (2) fine-tuning the classification layer. In the following, we mainly describe the training of the feature extractor and the prototypes. In this step, the classification layer weights are fixed by the predefined relationship between the class labels and prototypes, and the training is conducted to minimize the loss function composed by classification loss, cluster loss, and separation loss. Classification loss (typically, cross-entropy loss) is utilized to train the model so that it can solve the classification task. Cluster loss is used to link the image patch features and the prototypes within the same class, and the separation loss is vice versa. In ProtoPNet, the cluster loss and the separation loss are defined as

$$L_{clst} = \frac{1}{|\mathbb{B}|} \sum_{a\in\mathbb{B}} \min_{i\in\mathbb{P}_+^{y_a}} \min_{\boldsymbol{z}\in\mathbf{Z}_a} D(\boldsymbol{z}, \boldsymbol{p}_i),\ L_{sep} = -\frac{1}{|\mathbb{B}|} \sum_{a\in\mathbb{B}} \min_{i\in\mathbb{P}_-^{y_a}} \min_{\boldsymbol{z}\in\mathbf{Z}_a} D(\boldsymbol{z}, \boldsymbol{p}_i). \tag{16}$$

Here, we denote the function that calculates the (Euclidean) distances between the image patch features $\boldsymbol{z}$ and the prototype $\boldsymbol{p}$ as $D(\boldsymbol{z}, \boldsymbol{p})$. We also denote the index sets of the prototypes belonging to the class label $y_a$ as $\mathbb{P}_+^{y_a}$ and vice versa as $\mathbb{P}_-^{y_a}$. As explained above, ProtoPNet needs to define which prototype contributes to which class logits, and this definition enables each of the prototypes to represent useful image features for recognizing the class labels. This definition is also utilized in calculating the cluster loss. However, in the training for the similarity-based classifiers, this predefined relationship can no longer be used. Thus, ProtoPNet cannot directly utilize a similarity-based classifier for its final classifier. To address this problem, our proposed training process features a novel loss function that replaces the cluster loss in ProtoPNet (as described in Sec. 2.2.3).

After the training, each prototype is projected onto the nearest image patch features in the training set. This makes the prototypes transparent and enables us to interpret their meanings. Thus, the case-based interpretability is achieved by classifying samples based on the similarity of the input samples to the prototypes. Note that the interpretability of the classification layer is the only requirement for achieving case-based interpretability.

## B  DIFFERENTIABILITY OF PROPOSED CLUSTER LOSS

When calculating the cluster loss, we use the argmax function when sampling the prototypes (Eq. 2) to strictly set the affiliation of the prototypes that should not be included to zero. Therefore, the

affiliation of the prototypes to each class $P(y, p_i)$ is not differentiable with respect to the model parameter. In addition, since it consumes a lot of memory to obtain the gradient through the Sinkhorn-Knopp algorithm with the automatic differentiation, we treat $T_{a,i}^{y*}$ as the constant and use the gradient through $\hat{C}_{a,i}$ to train the model parameters. These problems for obtaining the gradients of $T_{a,i}^{y*}$ with respect to the model parameters can be solved by replacing the argmax function with softmax and providing the explicit derivative of the Sinkhorn-Knopp algorithm (Eisenberger, 2022). Validating these solution is outside the scope of this paper and is left for future work.

## C  Implementation Details

We conducted our experiments based on the implementation of Roth (2020) and used VGG16, VGG19, Resnet34, Resnet50, Resnet152, Densenet121, and Densenet161 for our model backbone in this paper. Following previous works (Nauta, 2021; Donnelly, 2022; Rymarczyk, 2022), we utilized the models pretrained on i-Naturerist 2017 when conducting experiments on CUB200-2011 and using ResNet50 for the model backbone. We also used the models pretrained on ImageNet for the other experimental settings. The size of the input images was transformed into 224×224, so the resolution of the feature map output from the model backbone was 7×7. Following previous studies (Rymarczyk, 2022; Nauta, 2021), we reduced the number of channels in the feature map to 128 for the experiments on Stanford Cars and to 256 for the other datasets by applying a 1×1 convolutional layer. After the channel reduction, we calculated the similarity between the prototypes and the image patch features contained in the feature map, and the maximum values of the similarity among the feature map were output as the similarities of the input samples to the prototypes. These similarities were then used to calculate $L_{task}$ during the training and to retrieve KNN samples during the inference. We also applied Global Average Pooling (GAP) and a linear layer to the feature map output by the model backbone and obtained the feature vectors, where the number of feature vector channels was set to the same as that of the prototypes. Note that these feature vectors were not used during the inference, though we did use them during the training to calculate $L_{aux}$. When cropped images were used, we followed previous work (Rymarczyk, 2022) and set the number of prototypes to 195 for Stanford Cars and to 202 for CUB200-2011. We also set the number of prototypes to 512 on all datasets when we used the full images.

During the training, we used the Adam optimizer and set the learning rate to 1e-5 for the model backbone and to 1e-3 for the other layers. For data augmentation, we used RandomPerspective, ColorJitter, RandomHorizontalFlip, RandomAffine, and RandomCrop following ProtoTree (Nauta, 2021). When testing with cropped images, we simply resized them to 224×224, and when using full images, we added RandomResizedCrop and set the minimum scale parameter to 0.6 during the training. We also resized the short side of the images to 256 and then cropped the center to resize them to 224×224 when testing with the full images. Further, to reduce the complexity during training, we first initialized the prototypes so that they were uniformly placed on the hyper-sphere and fixed them (Mettes, 2019), which means they were updated only when we projected them onto certain image patch features in the training set after initialization. The number of epochs was set to 140, 140, and 60 for the experiments on Stanford Cars, CUB200-2011, and Stanford Dogs, respectively. We constructed the minibatch so that it contained 56 classes and two images per class. When we used DenseNet161 for our model backbone, due to limitations on the GPU memory, the minibatch contained 42 classes and two images per class. Note that we report the average experimental results with three different seeds in this paper.

## D  Further Experimental Results

In this section, we explain the experimental results we omitted from the main paper due to space limitations.

### D.1  Ablation Study On The Cluster Loss

In this subsection, we first describe the experimental settings of the ablation study on the proposed cluster loss and then present the results. The difference of each setting here can be described as the difference in how much the image patch features are linked to each of the prototypes, i.e., the

difference in $T_{a,i}^{y*}$ in Eq. 9 or Eq. 11. This is because the image patch features nearest to the prototypes are linked to the prototypes in all of the settings. Table 6 summarizes the settings in the ablation study that are relevant to the difference in $T_{a,i}^{y*}$, as described below.

First, 'Just take the most similar' is adopted for the baseline, which simply links the image patches to the nearest prototypes. This setting is equivalent to setting $T_{a,i}^{y*}$ to $\mathbf{1}(i = \arg\min_j (\min_{\boldsymbol{z} \in \mathbf{Z}_a} \|\boldsymbol{z} - \boldsymbol{p}_j\|))$. In '+ Sampling', the image patch features are linked to the prototypes by using the estimated attribution of the prototypes to each class. Here, the attributions are estimated on the basis of the similarity difference of each sample in the minibatch to the prototypes. Specifically, we use the average of the sampling (Eq. 4) results within the samples that have the same class labels. Thus, in the setting of '+ Sampling', $T_{a,i}^{y*}$ is defined as $\bar{E}(y_a, p_i)/\sum_j \bar{E}(y_a, p_i)$ by using $\bar{E}(y, p_i)$ defined in Eq. 4. Here, we divide by $\sum_j \bar{E}(y_a, p_i)$ so that the cluster loss weight added to each sample is the same. In '+ De-bias', we use the de-biased attribution (Eq. 5) instead of the average of the sampling results, i.e., we define $T_{a,i}^{y*}$ as $\hat{E}(y_a, p_i)/\sum_j \hat{E}(y_a, p_i)$, where $\hat{E}(y, p_i)$ is defined in Eq. 5. Next, in '+ Linear assignment', we modify the attribution of the prototypes to each sample to account for the absence of the prototypes due to differences in the object views. Note that we modified the attribution by solving the linear assignment problem, so in the setting of '+ Linear assignment', we define $T_{a,i}^{y*}$ as in Eq. 10. Finally, in ' + Maximize the gap (full)', we redefine the cluster loss from Eq. 9 to Eq. 11. Note that the definition of $T_{a,i}^{y*}$ is the same as in '+ Linear Assignment'.

The results of the ablation study are shown in Table 7. We adopted ResNet50 for the model backbone and used the full images of CUB200-2011 in this experiment. As discussed in Sec. C, we fixed the prototypes on the hypersphere and thus the accuracy does not vary among the experimental settings. However, without debiasing, i.e., in the 'Just take the most similar' and '+ Sampling' settings, image patch features are linked to only a few of the prototypes by the cluster loss, and as a result, they are not linked to the other prototypes and are not distributed over the hypersphere, which means the prototypes after the projection are also not distributed on the hypersphere. This can be confirmed by the fact that the variance of the prototypes on the hypersphere in these case is much smaller compared to the other settings. Here, we defined the variance of the prototypes on the hypersphere as

$$1 - \left\| \frac{1}{|\mathbb{P}|} \sum_{i \in \mathbb{P}} \boldsymbol{p}_i \right\|_2 . \tag{17}$$

Since we initialized the prototypes so that their variance on the hypersphere would be maximized, the small value of the variance observed here suggests the destruction of the original learned structures by projecting the prototypes onto the image patch features.

We can see the accuracy is degraded when we adopted '+ Sampling' and '+ De-bias', which means the inconsistency of attachment between the image patch features and the prototypes in these settings. However, the accuracy is recovered while maintaining the variance high when we adopt '+ Linear assignment', which confirms that the image patch features were properly linked to the prototypes by modifying their attributions to account for the differences in object views by solving the linear assignment problem. Finally, in '+ Maximize the gap (full)', we can see that the variance further increased from that in '+ Linear assignment'. The reason for this increase may be that maximizing the gap also acted as a separation loss and thereby led to the dispersal of image patch features over the hypersphere. Our findings here demonstrate the effectiveness of each component of the proposed cluster loss.

Table 6: Summary of experimental settings of ablation study on cluster loss.

| | $T_{a,i}^{y*}$ |
|---|---|
| Just take the most similar | $\mathbf{1}(i = \arg\min_j (\min_{\boldsymbol{z} \in \mathbf{Z}_a} \|\boldsymbol{z} - \boldsymbol{p}_j\|))$ |
| + Sampling | $\bar{E}(y_a, p_i)/\sum_j \bar{E}(y_a, p_i)$ |
| + Debias | $\hat{E}(y_a, p_i)/\sum_j \hat{E}(y_a, p_i)$ |
| + Linear assignment | $\arg\min_{T_{a,i}^y} \sum_{a,i} T_{a,i}^y C_{a,i} + \frac{1}{\lambda} T_{a,i}^y \log T_{a,i}^y$ |
| | $s.t. \sum_i T_{a,i}^y = \mathbf{1}(y_a = y), \ \sum_a T_{a,i}^y = P(y, \boldsymbol{p}_i) \cdot \sum_b \mathbf{1}(y_b = y)$ |

Table 7: Results of ablation study on cluster loss with CUB200-2011. We used ResNet50 for our model backbone and Margin loss for $L_{aux}$.

|  | Top-1 Accuracy ↑ | | | Variance of the prototypes ↑ |
|---|---|---|---|---|
|  | k=1 | k=3 | k=5 |  |
| Just take the most similar | 87.0% | 87.1% | 87.3% | 0.6624 |
| + Sampling | 86.2% | 86.6% | 86.7% | 0.6646 |
| + Debias | 86.7% | 87.0% | 87.0% | 0.7267 |
| + Linear assignment | 86.9% | 87.1% | 87.2% | 0.7211 |
| + Maximize the gap (full) | 87.0% | 87.1% | 87.2% | 0.7489 |

## D.2 FURTHER EXAMPLES OF THE INTERPRETATION OF THE PROPOSED METHOD

Examples of the interpretation of our method are shown in Figs. 4 and 5. Here, in both figures, we used randomly selected samples and negative classes.

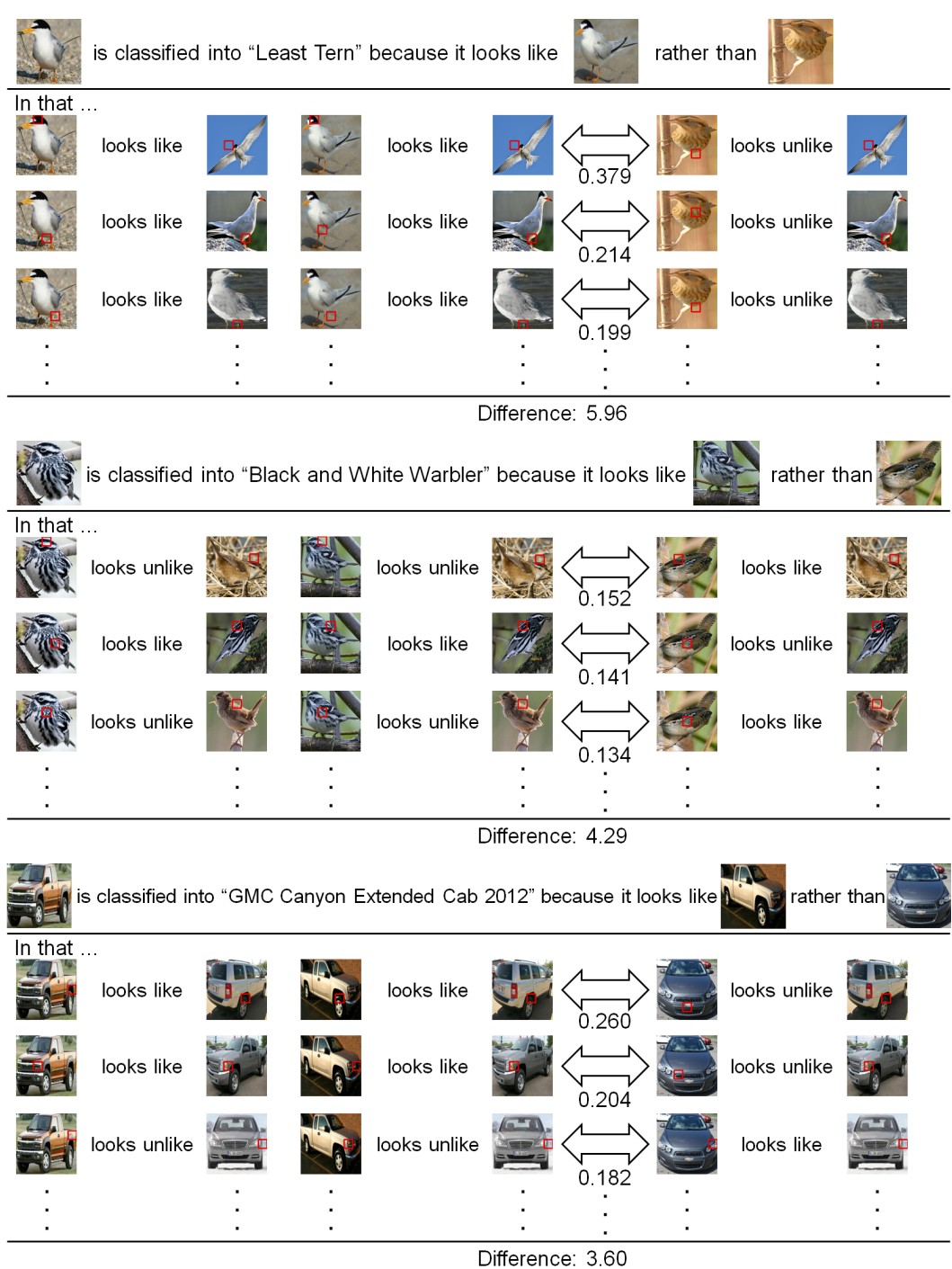

Figure 4: Examples of the interpretation of the proposed method. Red bounding boxes surround the areas where the prototypes are activated the most.

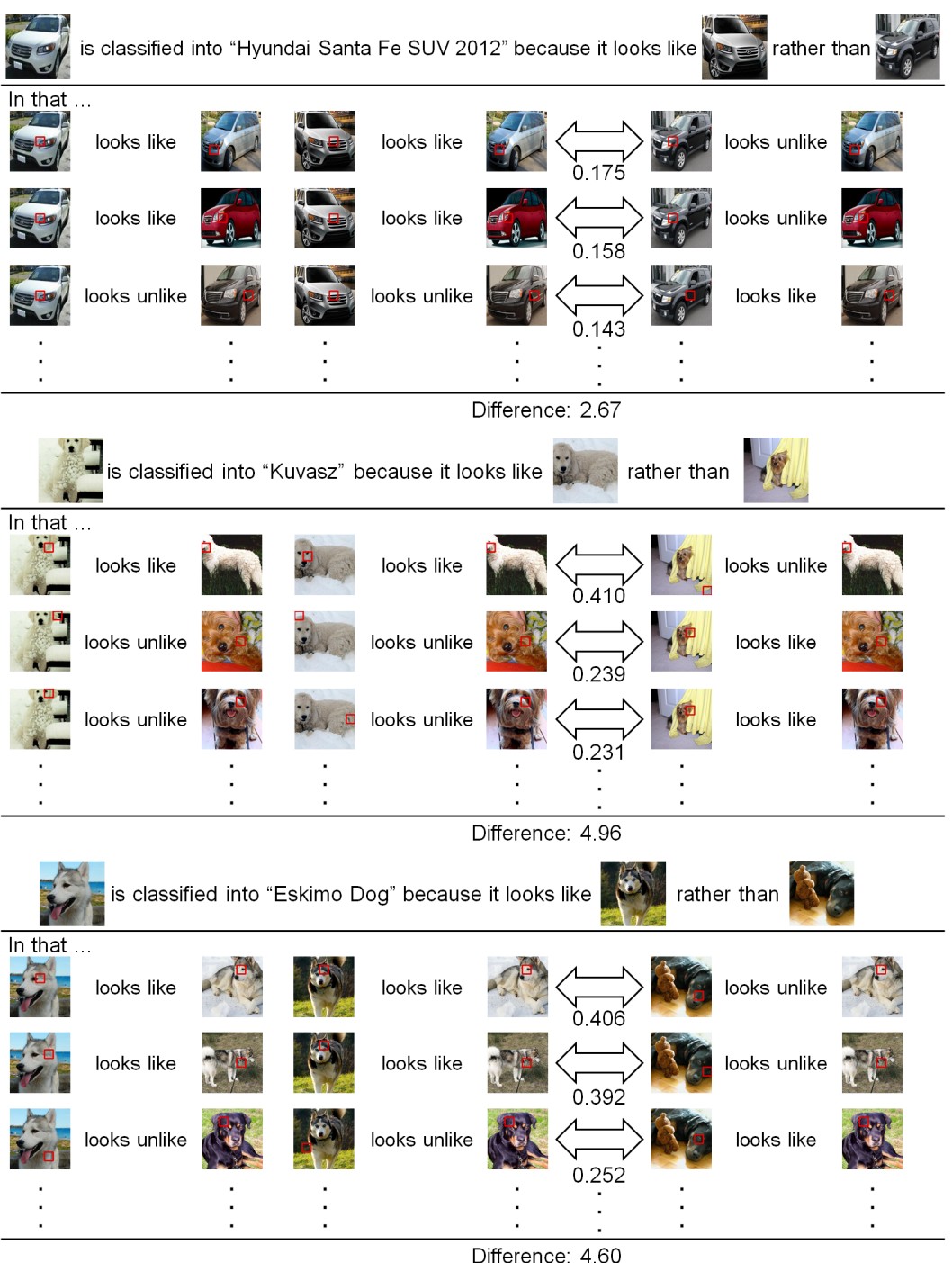

Figure 5: Examples of the interpretation of the proposed method. Red bounding boxes surround the areas where the prototypes are activated the most.

