# OpenReview forum: "This Looks Like It Rather Than That: ProtoKNN For Similarity-Based Classifiers"
_ICLR.cc/2023/Conference — ICLR 2023 poster_

### Official Review · Reviewer_o3Fc · 2022-10-25

**Confidence:** 3
**Correctness:** 3
**Technical Novelty And Significance:** 3
**Empirical Novelty And Significance:** 3
**Recommendation:** 6

**Clarity, Quality, Novelty And Reproducibility:**

In general, the paper is of a reasonable quality, although the clarity of the paper can certainly be improved. There is some novel components of the paper, e.g., modified cluster loss, but the idea of replacing the fully-connected classification layer with an kNN is only incremental in novelty. I do not have concerns with regard to reproducibility.

**Details Of Ethics Concerns:**

There are no ethical concerns.

**Strength And Weaknesses:**

Strengths:

- Interpretability: The proposed model is interpretable. Its predictions can be explained in terms of training images with similar prototypical activations. In particular, the proposed model can explain why an input image is similar to another image of the predicted class, but is dissimilar to an image not of the predicted class.
- Accuracy: The proposed model improves accuracy over existing variants of ProtoPNet models.

Weaknesses:

- Technical soundness: In equation (3), how do you choose the hard-positive and the hard-negative samples for computing the task loss?
- Clarity: The description of cluster loss is difficult to follow. In addition, Figure 3 is confusing -- I had a hard time trying to understand it.

**Summary Of The Paper:**

In this paper, the authors proposed a ProtoKNN model, which uses k-nearest neighbor (kNN) on prototype similarities to classify an input image. In particular, the proposed model uses a ProtoPNet backbone (with cosine similarity), and classifies an input image by comparing its prototype similarities with those of the training images. After finding the top k training images whose prototype activations are similar to the prototype activations of the input image, the model uses the majority class label of those k images as the predicted class of the input image. To train their ProtoKNN model, the authors used a loss function consisting of: (1) a task loss, which encourages an input image to have similar prototype activations as another image of the same class, and to have different prototype activations from an image of a different class; (2) a modified cluster loss, which encourages each training input to have at least one patch close to a prototype that is affiliated with its own class; and (3) an auxiliary loss (e.g., margin loss, proxy anchor loss) from deep metric learning. The authors compared their ProtoKNN with ProtoPNet, Deformable ProtoPNet, and other variants of ProtoPNet models, and claimed that their model is able to achieve a higher accuracy even without ensembling. In addition, the authors also demonstrated the explanations generated by their model (Figure 3).

**Summary Of The Review:**

Based on the strengths and weaknesses discussed above, I believe that the paper has some value, but needs improvement especially in terms of clarity.

---

### Official Review · Reviewer_RzJT · 2022-10-25

**Confidence:** 3
**Correctness:** 3
**Technical Novelty And Significance:** 3
**Empirical Novelty And Significance:** 2
**Recommendation:** 6

**Clarity, Quality, Novelty And Reproducibility:**

-There is novelty in extending the original ProtoPNet architecture to work with KNN.

-The methodology is not properly described. The intuition behind many of the equations are not explained. (see my notes above)

-My main concern is that it is not clear which drawback of the previous variants of the ProtoPNet is addressed by using KNN. From the provided results, it does not seem to be about an improved accuracy or efficiency.

**Strength And Weaknesses:**

+The paper adapted the general architecture of ProtoPNet (chen 2019) to work with a KNN rule. Previously we have seen ProtoTree which uses a decision tree instead of a linear layer. The current paper uses KNN.

-From algorithmic perspective, what is the advantage of using ProtoKNN over other methods that are not based on KNN? The authors listed extending the ProtoPnet to use KNN as a contribution. But, what are the advantages of using KNN with ProtoPNet compared to other variants? In the introduction, authors briefly mentioned "situations where unknown classes or anomalies exist. But the rest of the paper does not provide any further explanation or any experiment to support the effectiveness of the proposed method in those situations.

-The method is not properly described. See my specific questions below.

-A lot of pairwise distances between images patch features are required. An analysis of the training and inference time is required. The performance of the proposed method is on par with Deformable ProtoPNet (Donnelly, 2022), so the training time becomes important. The authors provided the results of ensemble of Deformable ProtoPNet and all other comparison methods but not for their own method. Is it because of the time complexity of training the proposed method that prohibited doing so?

-The authors examined K=1, 3 and 5 in the KNN. In most experiments K=5 has performed better than others. This suggests that larger values of K should also be examined.

-What is the summation in the denominator of eq. 7?

-What is the summation in the denominator of eq. 8?
Neither are explained other than vaguely stating they are for “debiasing” and “normalizing”.

-Eq. 9 is not properly described. What each summation is for?

-The auxiliary losses are not described in the methodology. Their definitions are provided in the ablation analysis in Sec 3.2. I suggest moving their description to where they are introduced.

-Comparison methods have been used along with different backbones. ProtoPNet and Deformable ProtoPNet are used with VGG19, Res152 and DenseNet161 backbones, but ProtoTree and ProtoPool are used with Res50 backbone. Therefore it is not possible to compare the results. Why? Also why the results of the proposed method for Res152 is inferior to R50?

-The results of the ablation analysis in Table 7 show that the sampling, debiasing and linear assignment steps have no significant effect on the accuracy. The provided justification is based on variance of prototypes. Why increasing variance from 0.66 to 0.74 is important? No qualitative results are provided to shed light on this.


**Summary Of The Paper:**

This work studies the effectiveness of KNN as a similarity-based classifier as opposed to the linear classifier of the original ProtoPNet. The paper also proposes a loss function to be used instead of the clustering term in the loss function of the original ProtoPNet. Another difference is that the paper does not use the separation loss. This allows the prototypes to be common among different classes which is previously addressed in ProtoPool (Rymarczyk 2022). They also use some additional losses from previous work empirically they showed that when added to the mix, they may help in some settings.  The performance of the proposed method is examined on two datasets and the results are competitive with the previous variants of ProtoPNet.

**Summary Of The Review:**

I have major concerns and I hope the authors can provide clarifications.

---

### Official Review · Reviewer_94zB · 2022-10-28

**Confidence:** 3
**Correctness:** 3
**Technical Novelty And Significance:** 2
**Empirical Novelty And Significance:** Not applicable
**Recommendation:** 6

**Clarity, Quality, Novelty And Reproducibility:**

Clarity-- Fair:
Some of the text is hard to follow, please see the clarifications above.

Quality --Medium:
ProtoKNN outperforms other baselines. However, due to the use of KNN I believe the use of ProtoKNN would be much more expensive compared to other baselines.

Novelty--Medium:
ProtoKNN is an extension of ProtoPNet to use KNN classifiers rather than linear models to do so  ProtoKNN dramatically changes the original proposed loss.


Reproducibility--Fair:
Code was not provided but some implementation details were available in the appendix.

**Strength And Weaknesses:**

Strength:
--

- Empirical evidence shows that ProtoKNN outperforms other baselines on multiple datasets and different architectures.
- Counterfactual explanations produced by ProtoKNN show both positive and negative examples.

------------------

Weakness:
--
- Training ProtoKNN is non-triaval.
- Method is quite expensive in inference time, especially for large datasets (ignoring any memory issues).
- No discussion of the computational cost  (training time etc..) was included.
---------------------
Clarifications:
-
   - It is not clear why using $L_{task}$ is better than using cross-entropy.
   - I am a bit confused about how the class label is chosen, the euclidean distance between samples is calculated between the raw samples or the prototypes?
  - I really appreciate figure 2 which tries to summarize the cluster loss, however, it is not very clear what do the color bars stand for.
  - Since  $L_{aux}$ is used in the main loss function maybe include the actual loss equation in the main text.



**Summary Of The Paper:**

The paper proposes an extension to ProtoPNet that utilizes KNN classifiers. To do this the paper replaces ProtoPNet loss with 3 losses:

a) A classification loss $L_{task}$ eq 3 which replaces the regular cross-entropy loss.

b) Cluster loss $L_{clst}$ summarized in fig 2; which first calculates the affiliation of the prototypes to each class than links the prototypes and image patch features based on the affiliation.

c) An auxiliary loss  $L_{aux}$ proposed in the context of deep metric learning by Wu,2017.

The total loss is the sum of all three losses with hyperparameters.

For label prediction top K samples from the training set based on euclidean distance and the majority vote is used.


**Summary Of The Review:**

Overall empirical results show that ProtoKNN produces better results from baselines and the explanations produced by ProtoKNN are very intuitive. However, my main concern is the cost of ProtoKNN and some clarity issues in the writing.

---

### Official Review · Reviewer_g4E3 · 2022-10-31

**Confidence:** 3
**Correctness:** 4
**Technical Novelty And Significance:** 3
**Empirical Novelty And Significance:** 2
**Recommendation:** 6

**Clarity, Quality, Novelty And Reproducibility:**

Clarity & Quality: the paper can be presented better.

Novelty: This paper is somehow novel.

Reproducibility: The results should not be hard to reproduce.

**Strength And Weaknesses:**

Strengths:
- The paper studies an important problem and addresses an issue of an existing popular approach. I generally appreciate the efforts of this paper.
- The motivation of this paper is clear and the paper is well-positioned; the related works have been discussed well.
- The method makes sense to me. The proposed cluster loss is novel and interesting.
- The experimental results are strong.
- The paper has discussed the limitations (e.g., memory efficiency, differentiability) fairly well.


Weaknesses:
- The paper can be presented more clearly. (1) The high-level idea of the method should be discussed in the introduction, while the current introduction only talks about motivation of the research; (2) section 2.3.2 is lengthy and not easy to follow; (3) figure 1 and figure 2 are hard to follow and there are nearly no captions for the figures; (4) the paper does not talk about the auxiliary loss, which can really confuse readers; (5) figure 3 is not easy to understand.
- Although in terms of experimental results, the paper is similar or stronger than previous approaches, I think the experiment section can be made more informative and insightful. For example, there is no analysis on the proposed cluster loss; also, I expect to see experiments where unknown class samples exist, such that similarity-based classifiers can be fully utilized.


**Summary Of The Paper:**

This paper aims at training deep learning models with interpretability. The paper first studies the existing ProtoPNet models, which calculate the similarity of the input to the prototypes and classifies samples on the basis of the similarity. The paper points out the issue of the vanilla ProtoPNet model -- ProtoPNet is not able to utilize any classifiers other than the linear classifier. To address this issue, the paper extends ProtoPNet and proposes ProtoKNN. The proposed ProtoKNN method does not require predefined relationship between classes and prototypes during training.The paper examines the effectiveness of similarity-based classifiers in the “this looks like that” framework by using ProtoKNN. The experiments on several open datasets show ProtoKNN achieves better or similar results compared to previous approaches.


**Summary Of The Review:**

In summary, I think this paper has clear strengths and weaknesses. I would like to assess it as a paper around the acceptance threshold.

---

### Decision · Program_Chairs · 2023-01-20

**Decision:**

Accept: poster

**Justification For Why Not Higher Score:**

As evoked during the discussion with reviewers', the paper still lacks to give more arguments on the value brought by using kNN. While being positives, the reviewers were not that impressed.

**Justification For Why Not Lower Score:**

The reviewers were positives and many of them improved their scores to reach 6. Even if the paper has weaknesses, the consensus was positive for a weak accept among reviewers. SAC has been informed about the proposition.


**Metareview: Summary, Strengths And Weaknesses:**

This paper presents an extension of ProtoPNet, an interpretable image classification model with deep learning to work with KNN using similarities to nearest prototypes to classify images, ProtoPnet being used as a backbone. The method has the advantage to be useful in domains where instances with unknown classes are present. The paper shows that the proposed contribution ProtoKNN can improve accuracy while maintaining interpretability.

Strengths:
-Interesting idea with novel loss
-Accuracy improved with respect to state of the art
-Interpretability of the model (but that come from ProtoPnet)

Weaknesses:
-the presentation can be improved
-The computational cost of the method is not analyzed and seems important.
-Situations where the method is more appropriate than others still needs deeper analysis.


During rebuttal, authors' have provided multiple answers to issues raised by the reviewers. Authors have improved the contribution in many aspects. Overall, many concerns have been addressed and there is a global positive consensus for this paper.
While there are still some issues, I recommend a weak accept.


**Note From Pc:**

if the above contains the word "oral" or "spotlight" please see: "oral" presentation means -> notable-top-5% and "spotlight" means -> notable-top-25%. As stated in our emails, we are disassociating presentation type from AC recommendations